# Optimizing Planting Density in Alpine Mountain Strawberry Cultivation in Martell Valley, Italy

Sebastian Soppelsa * , Michael Gasser  and Massimo Zago

Berries and Stonefruits Research Group, Laimburg Research Centre, 39040 Ora-Auer, Italy;
michael.gasser@laimburg.it (M.G.); massimo.zago@laimburg.it (M.Z.)
* Correspondence: sebastian.soppelsa@laimburg.it

**Abstract:** Optimizing profitability is a challenge that strawberry farmers must face in order to remain competitive. Within this framework, plant density can play a central role. The aim of this two-year study was to investigate how planting density can induce variations in plant growth and yield performances in an alpine mountain strawberry cultivation (Martell Valley, South Tyrol, Italy), and consequently quantify the farm profit. Frigo strawberry plants cv. Elsanta were planted in soil on raised beds and subjected to five different planting density levels (30,000 and 45,000 as large spacing; 60,000 as middle spacing; 90,000 and 100,000 plants ha$^{-1}$ as narrow spacing, corresponding to a plant spacing of 28, 19, 14, 9, and 8.5 cm, respectively). Our findings indicate that the aboveground biomass in plants subjected to low planting density was significantly increased by +50% (end of first year) and even doubled in the second year in comparison with plants in high planting density. Those results were related to higher leaf photosynthetic rate (+12%), and the number of crowns and flower trusses per plant (+40% both) ($p < 0.05$). The low yield (about 300 g plant$^{-1}$) observed in the high planting density regime was attributable to smaller fruit size during the first cropping year and to both a reduced number of flowers per plant and fruit size during the second year ($p < 0.05$). Although the highest yield (more than 400 g plant$^{-1}$) was obtained with wide plant spacing, the greatest yield per hectare was achieved with high planting densities (28 t ha$^{-1}$ in comparison with 17 t ha$^{-1}$ with low plant density level). However, the farm profit must take into account the costs (especially related to the plant material and harvesting costs) that are higher under the high planting density compared with the other density regimes. Indeed, the maximum farm profit was reached with a density of 45,000 plants ha$^{-1}$ which corresponded to EUR 22,579 ha$^{-1}$ (over 2 years). Regarding fruit quality, fruits coming from the low plant density level showed a significantly higher color index (+15% more red color) than fruits from high plant density ($p < 0.05$). In conclusion, our results suggest that a middle planting density can be a fair compromise in terms of plant growth, yield, and farm profit.

**Keywords:** *Fragaria x ananassa*; plant spacing; altitude; flowering; fruit quality

## 1. Introduction

Strawberry is a herbaceous perennial plant belonging to genus *Fragaria* of the family *Rosaceae* [1]. There are around 24 species of Fragaria in the world, mostly concentrated in China, making it the country with the largest genetic resources of wild strawberry [2]. Nevertheless, the strawberry species cultivated today derives from a natural hybridization that occurred in European gardens around the mid-1700's, between two species native to America (the South American *F. chiloensis* and the North American *F. virginiana*) [3]. Shortly thereafter, that new hybrid species, *Fragaria* x *ananassa*, was destined to become a popular fruit crop with a significant economic value [3]. Being a plant with great environmental adaptability, it is geographically distributed in various parts of the world [1]. According to data from the Food and Agriculture Organization of the United Nations (FAO), the worldwide production of strawberry was around 8.8 million tons in 2020 [4]. China is the largest strawberry producer in the world (3.3 M tons), followed by United States of America

(1.1 M tons), Egypt (0.6 M tons), Mexico (0.6 M tons), and Turkey (0.5 M tons). The leading European country in strawberry production is represented by Spain (0.3 M tons) [4].

According to the Italian National Statistics Institute (data updated to 2021), the national production of strawberries is around 121 thousand tons of which 29 thousand in open fields (1871 ha) and 92 thousand in greenhouses (2631 ha) [5]. Although much of the production is concentrated in southern and central Italy, the significant contribution provided by the northern regions, such as Emilia-Romagna, Veneto, and regions located along the Alpine arc (Piedmont and Trentino-South Tyrol), must not be overlooked, and thanks to this variability of Italian environments, a fresh product is guaranteed throughout the year.

Several areas of strawberry cultivation can be identified in the province of Alto Adige/Südtirol/South Tyrol (in Italian, German, and English, respectively), from Martell Valley to Isarco Valley and Pusteria Valley, thus covering an area of about 100 ha [6]. More in detail, the Martell Valley on the southern side of Venosta Valley allows the cultivation of strawberry (and other berries) in an alpine mountain environment. The beginnings of the "heroic strawberry cultivation" in Martell Valley originated in the 60 s, when a group of farmers, with the help of the Department of Agriculture of the Autonomous Province of Bolzano/Bozen, identified the potential of the strawberry to be grown in this microclimate. The production of strawberries extends from 900 to 1800 m a.s.l., hence the name "Martell Strawberry Valley". Due to the altitude of the growing areas, Martell Valley is considered the highest cultivation area for strawberries in Europe. Late spring planting systems with cold-stored plants are currently the most adopted way by the farmers of the valley. These plants guarantee two crops, one in the year of planting, the other in the following year. The "Martell strawberry" ripens very slowly, in this way the fruits take on unique aromas and fragrances, and make themselves available from June to September, a period in which great national and European productions are absent [7].

Although the Italian strawberry acreage has drastically collapsed over the years (11,000 ha in 1989 to the current 4500 ha in 2021), yield per hectare, however, has resulted in an increase of 47% [5,8]. This improvement is attributable to two factors: breeding programs and growing systems. The first case results in the release of new, more productive cultivars with higher fruit quality parameters and plant resistance/tolerance to pathogens. In the second case, a traditional cultivation system (soil cultivation in open field or protected) has often been replaced by an advanced soilless system [9]. Nevertheless, some strawberry areas are linked to the tradition of the past with a soil cultivation in open field and with historical cultivars (e.g., cultivar Elsanta); this is the case of Martell Valley with very low yields per plant due to both the limiting environmental conditions and the lack of information on some correct agronomic practices (e.g., suitable planting density).

Plant density is simply expressed as the number of individuals per unit ground area [10]. According to several studies conducted primarily on herbaceous crops, plant morphology and productivity are influenced by the manipulation of plant density, more specifically synthesis of chlorophyll, photosynthesis, plant growth, floral induction, and flower formation are affected by different crop spacing [11–15]. The right crop density is certainly essential to obtain a maximum yield and income in strawberry cultivation [16]. Both low and high plant densities can reduce yield and total revenue. In other words, individual plants grown with a large spacing perform their best growth in terms of yield per plant but a low productivity per hectare [16]. On the contrary, as the distance between the plants decreases, a competitive relationship intensifies among individuals for limiting factors such as light, water, and nutrients, leading to a worsening of plant performances [17]. Irrigation and fertilization in open-field conditions are consolidated management practices to overcome or avoid abiotic stresses in relation to water-shortage or nutrient deficiency, respectively [18]. High plant density leads to mutual shading and self-shading of the leaves, thus hindering a correct interception of light [19]. Consequently, plants grown in that condition are subjected to morphological and anatomical changes, producing less biomass (i.e., leaves, roots), delaying flowering more than plants in full sunlight [20]. Looking at scientific literature, the interaction between reproductive phenology in strawberry plant

and some environmental parameters (e.g., light intensity, light quality/photoperiod, and temperature) is a topic of particular interest as evidenced by some studies [21–25].

An optimal plant density is calculated by identifying a density threshold beyond which the increase in individual plants does not lead to an increase in revenues [26]. In the study conducted by Wamser et al. [27], the fruit yield of tomato plants cultivated in humid subtropical climate in Calmon, State of Santa Catarina, Brazil (1208 m a.s.l.) was optimized with a plant density of 23,000 plants ha$^{-1}$, while increasing the number of individual plants increases yield but not the profit.

In another study, also conducted in Brazil on tomato cultivation, Carvalho et al. [28] found an optimal plant density of around 30,000 plants ha$^{-1}$ in Ipameri, State of Goiás (altitude 794 m).

Although many studies were conducted to determine an optimal plant density in several vegetable and fruit crops such as strawberry [14,29–34], the results that emerge from those research publications depend on some environmental and cultivation factors, and therefore they have a practical significance in the conditions in which the tests were carried out. A geographical climatic factor such as altitude that affects temperature and radiation has a fundamental role in changing plant responses (e.g., photosynthetic behavior, floral induction, fruit quality) [35–38].

As far as we know, no previous research has investigated the interaction between flowering/yield of strawberry plant and high altitude, combined with different plant densities. The present study aimed to investigate the effects of different plant densities on the growth, flowering, yield, fruit quality, and economic aspects of strawberry plants cv. Elsanta cultivated in a unique alpine mountain environment.

## 2. Materials and Methods

### 2.1. Field Management and Experimental Design

The experiment was conducted over two growing seasons (years 2020 and 2021) in an experimental strawberry field managed by the Laimburg Research Centre and located in the municipality of Martell (46°33′30.618″ N; 10°46′53.649″ E; 1.312 m a.s.l.) in South Tyrol, Italy. Martell Valley, a side valley of Venosta Valley included in the Stelvio National Park, is famous for berry production, in particular strawberry and a typical alpine mountain climate characterizes the valley. The soil properties of the 0–20 cm soil layer before planting in May 2020 were as follows: humic loamy sand, pH = 5.1, no free carbonate, organic carbon expressed as humus of 7.3%, phosphorus = 5.0 mg 100 g$^{-1}$, potassium = 8.0 mg 100 g$^{-1}$, and magnesium = 18.0 mg 100 g$^{-1}$. Meteorological trends during the growing seasons (from May to August 2020 and 2021) were recorded by iMETOS® weather station with the cloud platform "FieldClimate" (Pessl Instruments, Weiz, Austria) and data are reported in Table 1.

**Table 1.** Climatic conditions (monthly air temperatures, relative humidity, and rainfall) measured from May to August 2020 and 2021 during the first and second cropping year, respectively.

| | Air Temperature (°C) | | | Relative Humidity (%) | Rainfall (mm) |
|---|---|---|---|---|---|
| | Minimum Temperature | Maximum Temperature | Mean Temperature | | |
| **2020** | | | | | |
| May | 0.1 | 22.2 | 10.9 | 67.9 | 56.8 |
| June | 3.4 | 27.6 | 13.8 | 76.0 | 97.6 |
| July | 5.8 | 31.0 | 16.8 | 74.6 | 70.5 |
| August | 6.0 | 31.7 | 16.3 | 80.1 | 120.4 |
| **2021** | | | | | |
| May | 0.0 | 21.4 | 9.3 | 64.9 | 130.8 |
| June | 6.6 | 30.0 | 17.5 | 64.0 | 39.8 |
| July | 7.6 | 27.8 | 17.0 | 71.6 | 135.0 |
| August | 6.2 | 29.2 | 15.9 | 71.8 | 137.4 |

In our experimental test, frigo strawberry plants cv. Elsanta (heavy waiting bed (HWB) plants from the nursery: Neessen Aardbei and Aspergeplanten, Grashoek, Netherlands) were planted in soil conditions, precisely on raised beds with white plastic mulch films on the 31 May 2020 and subjected to five different plant density levels (30,000 and 45,000 as large spacing; 60,000 as middle spacing; 90,000 and 100,000 plants ha$^{-1}$ as narrow spacing, corresponding to a plant spacing of 28, 19, 14, 9, and 8.5 cm, respectively) (Figure 1). Plants were managed in the same way in terms of watering, fertilization, and pest control. The field received standard horticultural cares in accordance with the regulation governing integrated production. The experiment setup was organized as a completely randomized block design with 4 replicates composed of 40 plants per experimental unit (i.e., 120 plants per plant density level).

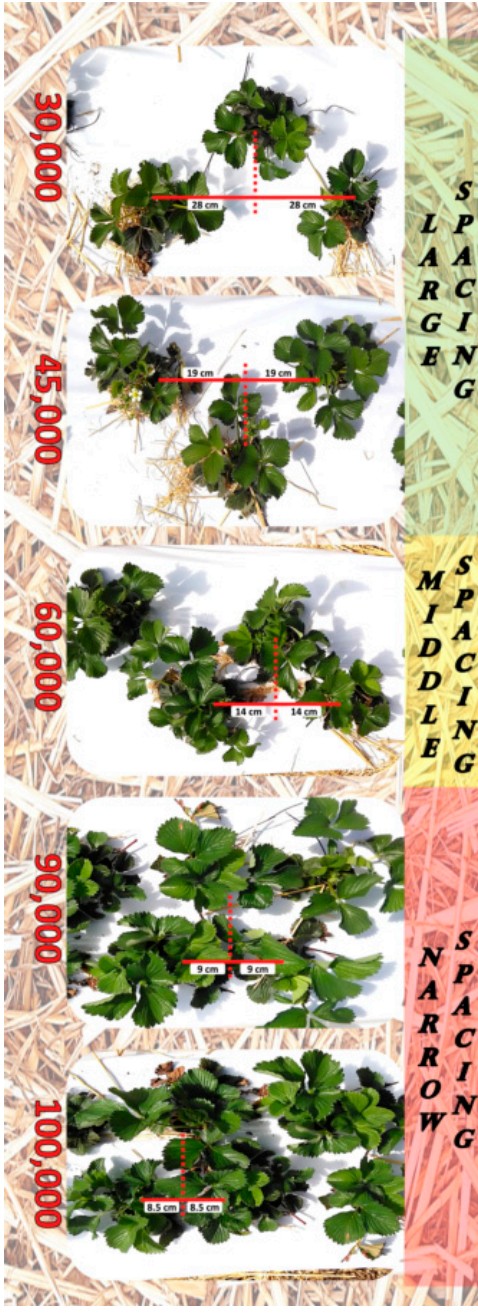

**Figure 1.** Strawberry plants cv. Elsanta planted in a double row on raised beds and subjected to different plant spacing.

## 2.2. Evaluated Parameters

### 2.2.1. Morphological and Gas Exchange Parameters

Main characteristics related to plant flowering were evaluated by counting the number of developed flower trusses per plant by destructively sampling ten randomly selected plants in each replicate after each harvest period. Thus, each flower truss was carefully assessed through a lens to determine the flower number (counted flower pedicel scars) per truss and per plant. Plant growth as affected by plant density level was determined by dissecting the same ten plants per plot previously mentioned. Then, each selected plant was separated into roots and aerial parts (leaves, crowns, flower trusses, and runners) and weighed fresh (g fresh weight (FW) plant$^{-1}$). The number of crowns per plant was evaluated by distinguishing between the main crown and branch crown. Afterwards, all plant organs were put in an oven (ED 56, Binder GmbH, Tuttlingen, Germany) at 65 °C until they reached a stable weight and the dry mass was recorded (g dry weight (DW) plant$^{-1}$). In the flowering stage, the net assimilation rate (A, $\mu$mol m$^{-2}$ s$^{-1}$) of leaves was evaluated using a portable infrared gas exchange analyzer (CIRAS-2, PP-Systems$^{\circledR}$, Hitchin, UK), attached to a PLC-6 cuvette having a measuring window of 2.5 cm$^2$. The $CO_2$ concentration (380 mmol mol$^{-1}$), PPFD (1500 $\mu$mol m$^{-2}$ s$^{-1}$), leaf temperature (25 °C), and air humidity (80%), were controlled by the device. Measurements were performed on a sun-exposed (clear and sunny days between 11:00 a.m. and 13:00 p.m.), young, fully expanded single leaf of four randomly selected plants per planting density.

### 2.2.2. Yield Parameters

Ripe strawberry fruits (uniformly red) were harvested every four days during the period from mid-July to mid-August 2020 (first harvest year) and throughout the month of July until the 7 August 2022 (second harvest year). From each experimental unit and at each picking time, the commercial production (healthy fruit with a diameter ≥22 mm) and the waste, represented by small fruit (diameter <22 mm), deformed and with the presence of rot, were weighed with a digital scale (Valor™ 2000, OHAUS Europe GmbH, Nänikon, Switzerland). The total production per plant (g fruit$^{-1}$) was calculated by dividing the harvested total fruit weight by the number of plants (considered 30 plants per experimental unit). The average fruit weight (g fruit$^{-1}$) was estimated by randomly sampling 10 fruits at each picking time.

### 2.2.3. Fruit Quality

Fruit quality was assessed on ten healthy strawberries per replicate which corresponds to 40 fruits analyzed per treatment. The fruits were sampled at two intermediate picking times for each harvest year. Flesh firmness was expressed with the Durofel index (DI) which represents the elasticity of the skin of the fruit (Agrosta$^{\circledR}$ Winterwood instrument, Agrosta Sàrl, Serqueux, France). The total soluble solids (°Brix) were determined with a refractometer (RFM840, Bellingham-Stanley Ltd., Kent, UK), whereas the titratable acidity (g L$^{-1}$ of citric acid) was measured with a titrator (Flash Automatic Titrator, Steroglass, Perugia, Italy) by titrating strawberry pulp to pH 8.2 using 0.1 M NaOH. The external fruit color was assessed with a colorimeter (CR-400, Konica Minolta, Tokyo, Japan) by measuring the same ten fruits at three different positions around the equatorial side of each fruit. The colorimetric coordinates (L*, a*, b*) were used to calculate the color index [CI = (1000 × a)/(L × b)] with higher CI value, indicating a more intense red color in the fruit [39].

### 2.2.4. Economic Analysis

A cost–benefit analysis was carried out for each plant density system. Profit is calculated by subtracting all farm's costs (variable and fixed) from the total revenue. Variable costs vary in relation to production volume, and in our case they referred to labor, plant material, mulch film, pesticides, fertilizers, and fuel. Instead, fixed costs remain the same regardless of production level and we considered the depreciation and maintenance quotas

of durable capital (tunnel structure, irrigation system, machinery, and buildings), administrative costs, and interest. Revenue is the total income generated from the sale of the strawberry fruits. Data are presented as total revenue, total costs, and farm profit for two consecutive years of cultivation according to planting density. Moreover, the profitability index, calculated by the ratio between gross income and total costs, is also reported in order to provide an indication of which option (i.e., planting density) is more profitable.

### 2.3. Statistical Analysis

Data normality was examined with the Shapiro–Wilk test, and homogeneity of variance was confirmed using Flinger–Killeen's test. A two-way ANOVA was performed on data collected from both years and mean separation of the dependent variables obtained with the LSD Fisher's test ($p < 0.05$). In case of significant interaction between "treatments" and "years", results were presented separately for the 2 years in dedicated tables or figures. A one-way ANOVA was performed on photosynthetic data coming from a single cropping year (2021). For non-normal data, Kruskal–Wallis test was applied. No statistical analysis was conducted for the economic part. All analyses were carried out in R v. 3.3.1. (R Development Core Team 2022). Values were expressed as mean ± standard deviation (SD).

## 3. Results

### 3.1. Morphological and Gas Exchange Parameters

We observed a worsening of individual plant biomass (roots and aerial part) by reducing the space between plants (Figures 2 and 3). Low plant density treatments showed plants with increased plant biomass (around +50%) in 2020 and doubled the biomass per plant in 2021 compared with values obtained in high plant densities. This result is not attributable to root biomass (no significant differences among treatments) but to the development of the aboveground part intended as leaves, crowns, flower trusses, and stolons.

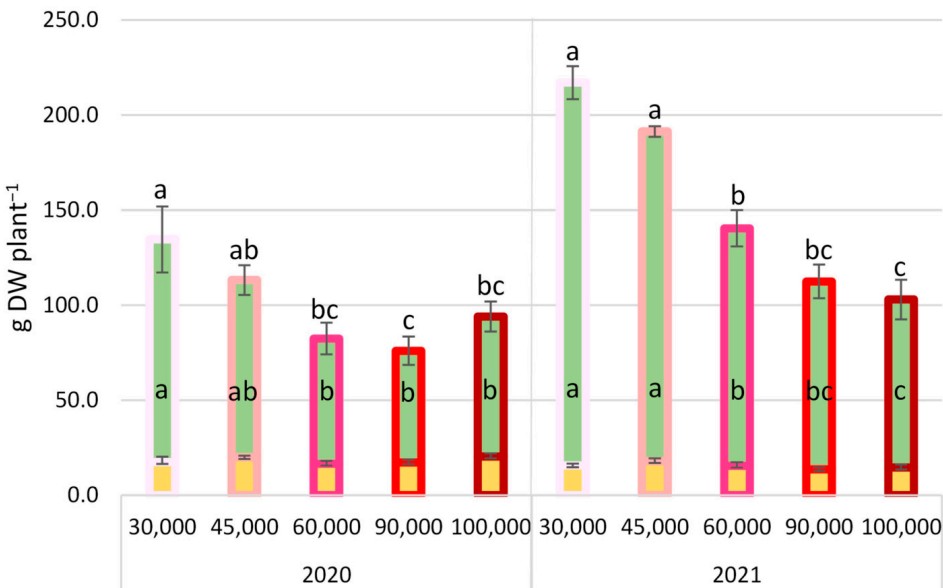

**Figure 2.** Total biomass (dry weight (DW)) composed of aboveground (green fill) and root biomass (yellow fill) at the end of first (2020) and second (2021) cropping year, as affected by planting densities. Vertical bars indicate means ± SD (*n* = 4). Within each year, the letters on the top of the bar (total biomass) and the letters on a green background (aboveground biomass) indicate significant differences according to LSD Fisher's test; *p* < 0.05. Root biomass data were not statistically significant.

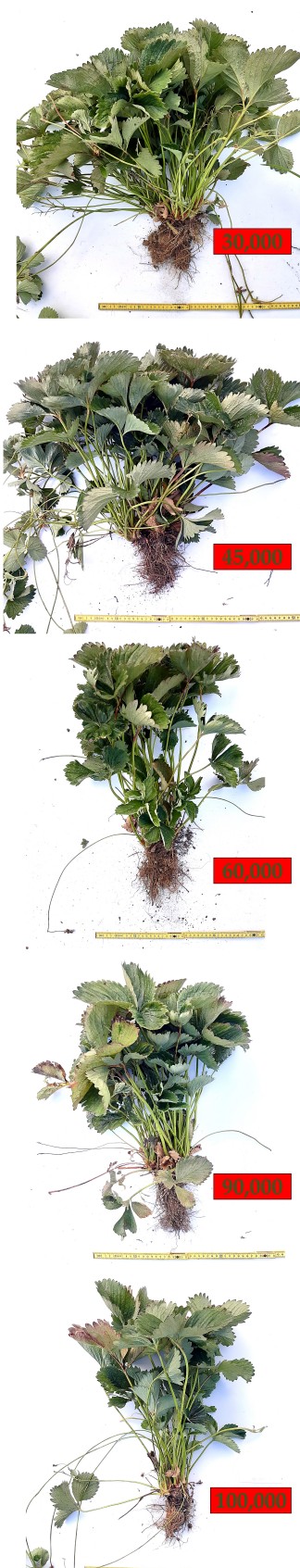

**Figure 3.** Strawberry plant biomass at the end of the second cropping year, as affected by planting densities.

The number of primary crowns and formed branch crowns (about three per plant) was statistically not affected by plant spacing treatments in 2020 (Figure 4). In the second year of growth (2021), plants at a larger plant spacing (30,000 or 45,000 plant ha$^{-1}$) showed more crowns than plants in 90,000 plant ha$^{-1}$ or 100,000 plant ha$^{-1}$ (7.5 and 5.5 crowns, respectively).

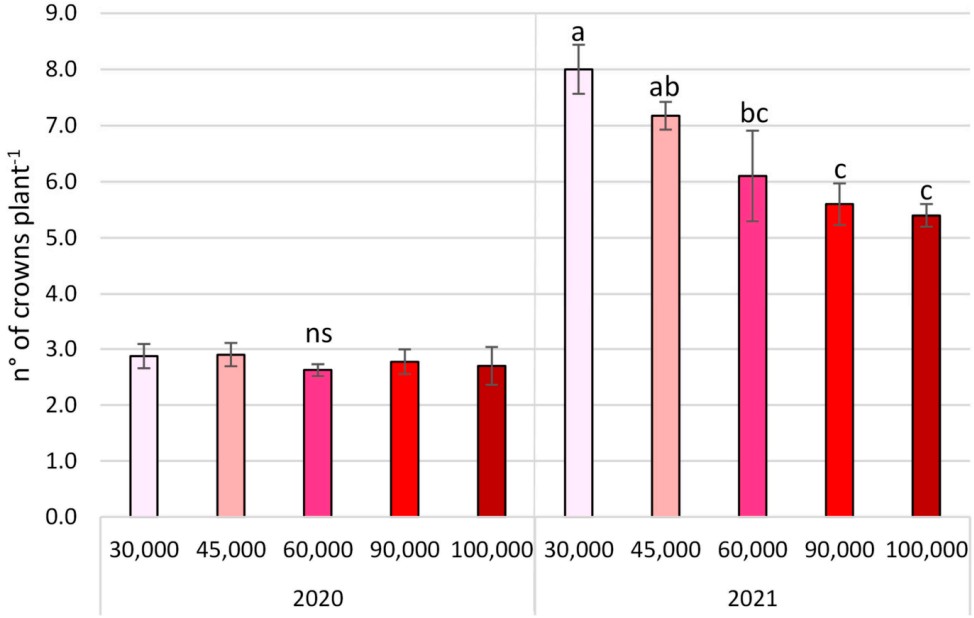

**Figure 4.** Number of total crowns (main and branch crowns) at the end of first (2020) and second (2021) cropping year, as affected by planting densities. Vertical bars indicate means ± SD (*n* = 4). Within each year, the letters indicate significant differences according to LSD Fisher's test; *p* < 0.05 (ns: not significant).

The net assimilation rate (A, μmol m$^{-2}$ s$^{-1}$) was evaluated only in year 2021 (Figure 5). A significantly higher leaf photosynthetic rate (+12%) was measured for leaves in plants subjected to large planting density.

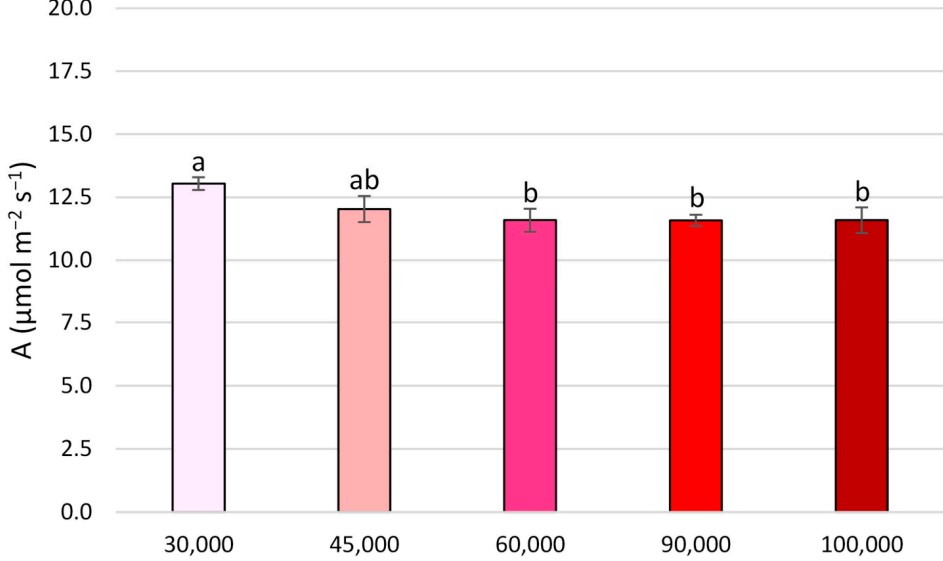

**Figure 5.** Photosynthetic rate in strawberry leaves during flowering, as affected by planting densities. Vertical bars indicate means ± SD (*n* = 4). The letters indicate significant differences according to LSD Fisher's test; *p* < 0.05 (ns: not significant).

　　　　Floral characteristics were affected by planting density, depending on the cropping year (Figure 6). As floral inductive conditions were the same during the nursery period, no significant differences were observed during the first year. A completely different situation in the second year highlighted the influence of plant density on flowering. Indeed, plants subjected to large spacing were characterized by more flower trusses per plant than plants cultivated in high density (8.4 and 5.9, respectively). The highest number of flowers per truss was identified at 45,000 plant ha$^{-1}$. More flower trusses and flowers per truss in larger spacing plants implied that the total number of flowers per plant appeared to be significantly greater in those plants compared with plants in high planting densities. The medium plant density (60,000) was significantly similar to narrow plant spacing.

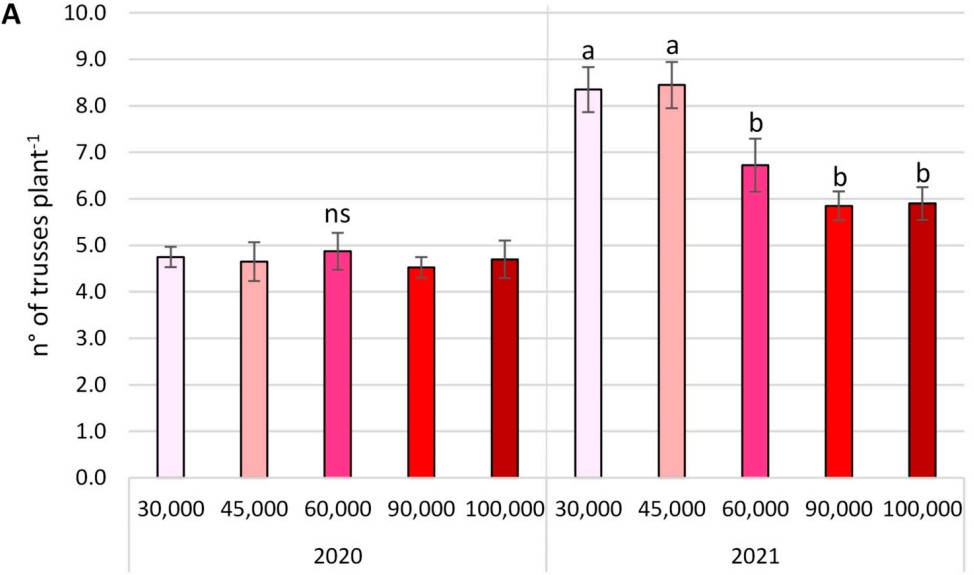

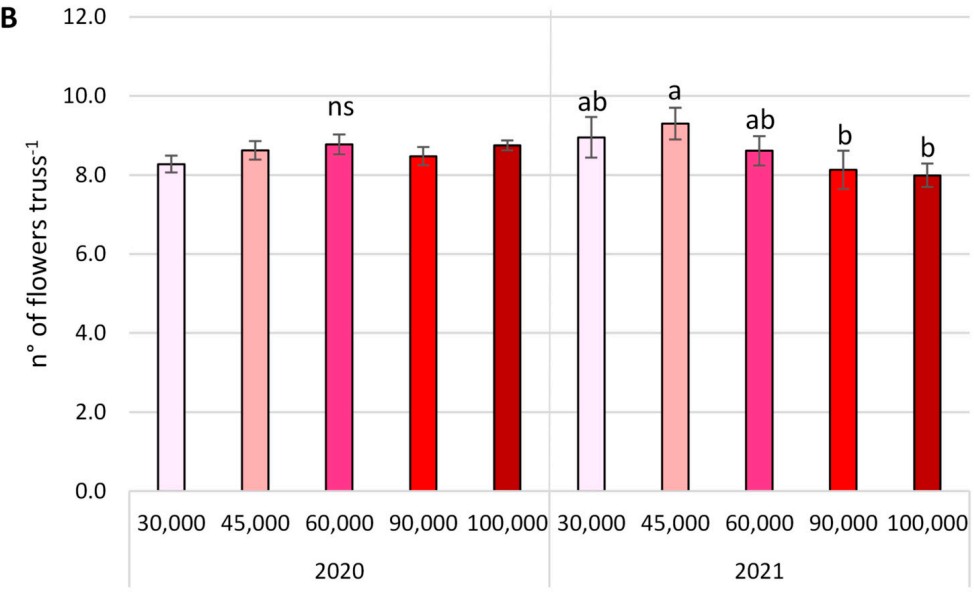

**Figure 6.** *Cont.*

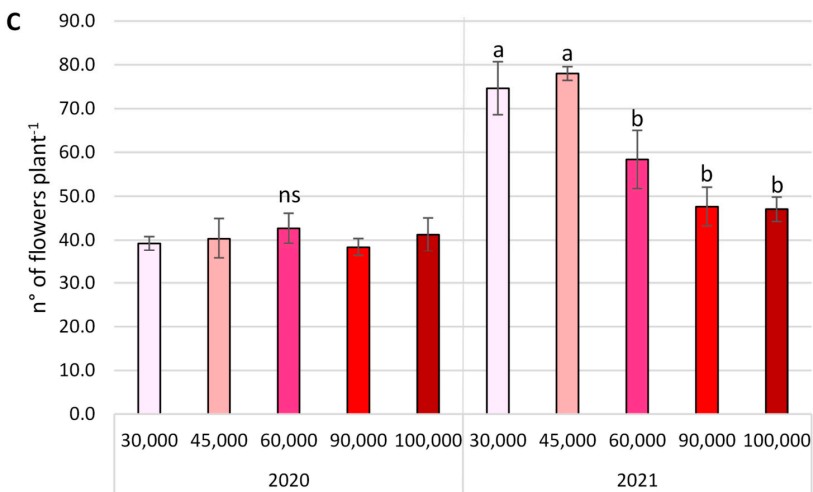

**Figure 6.** Reproductive characteristics of strawberry plants (flower trusses per plant—(**A**); flowers per truss—(**B**); total flowers per plant—(**C**) at the end of first (2020) and second (2021) cropping year, as affected by planting densities. Vertical bars indicate means ± SD (*n* = 4). Within each year, the letters indicate significant differences according to LSD Fisher's test; *p* < 0.05 (ns: not significant).

### 3.2. Yield Parameters

The strawberry production and its yield components are reported in Figures 7 and 8 and Table 2. Plants cultivated with wide-middle spacing were characterized by a significantly higher total yield per plant during the first (+36%) and second cropping year (+51%) than those in small spacing (Figure 7). Furthermore, there was a slight advance in fruit ripening in low planting density regimes, highlighting how this agronomic technique can influence the different phenological phases.

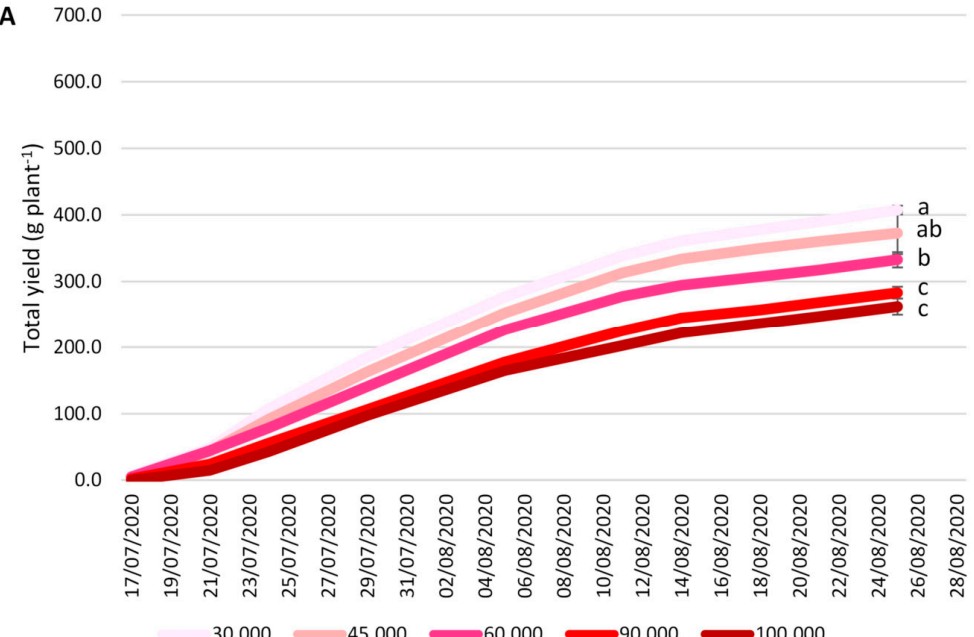

**Figure 7.** *Cont.*

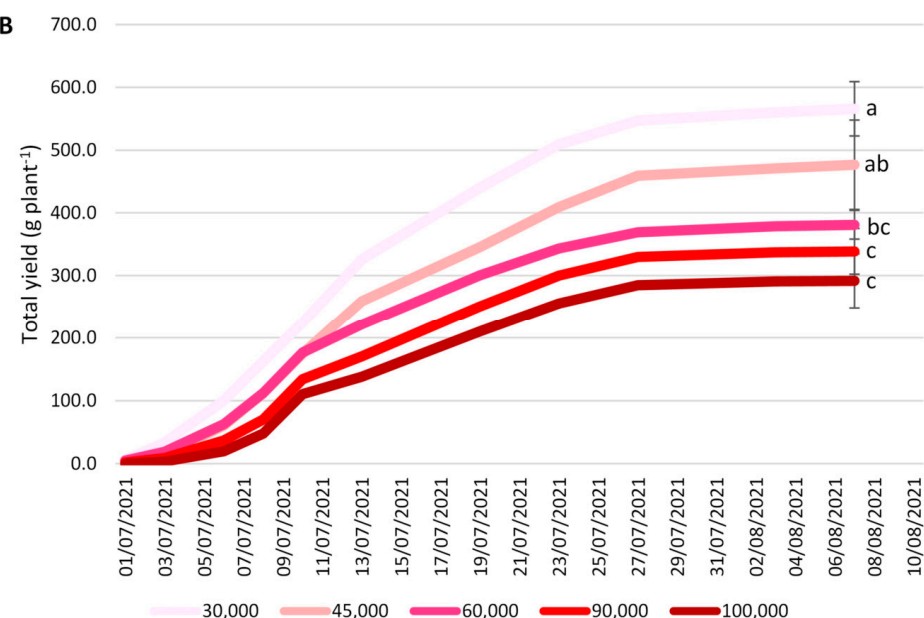

**Figure 7.** Cumulative total yield of strawberry plants in the first (2020—(**A**) and second (2021—(**B**) cropping year, as affected by planting densities. Vertical bars indicate means $\pm$ SD ($n = 4$). The letters indicate significant differences according to LSD Fisher's test; $p < 0.05$ (ns: not significant).

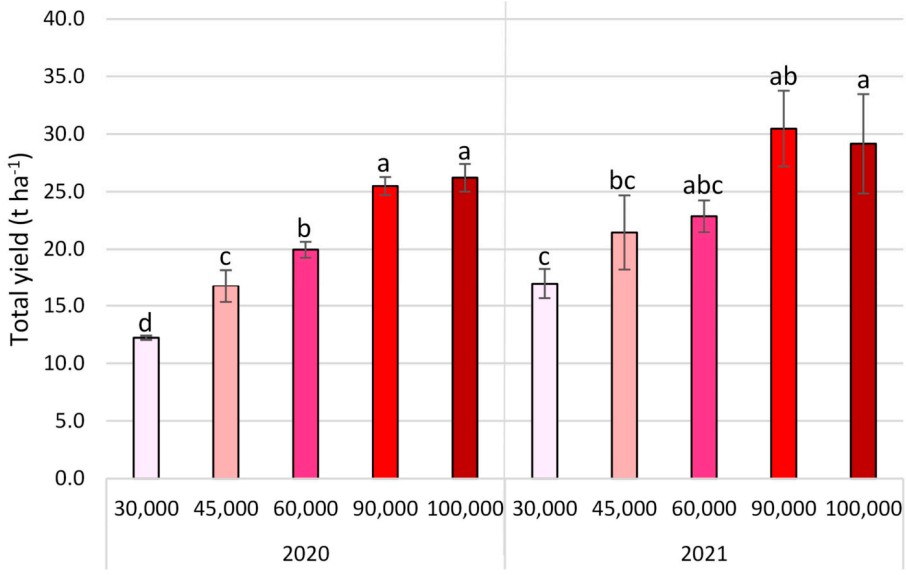

**Figure 8.** Total yield, expressed as tons per hectare, at the end of first (2020) and second (2021) cropping year, as affected by planting densities. Vertical bars indicate means $\pm$ SD ($n = 4$). Within each year, the letters indicate significant differences according to LSD Fisher's test; $p < 0.05$ (ns: not significant).

The analysis of yield components shows that the increased production per plant in 30,000 and 45,000 (and partially in 60,000) was due to the significantly highest quantity of first-class commercial berries (+65%) and misshaped fruits (+45%) compared with 90,000 and 100,000. No significant differences emerged for small and rotten fruits (Table 2).

As plant density increased (from 30,000 to 90,000), the productivity, expressed as tons per hectare, increased linearly (Figure 8). No significant difference was found between 90,000 and 100,000. Although the statistical differences were clear during the first cropping year, the productivity values for middle planting densities (i.e., 60,000) were not significantly different from the values displayed in low and high planting densities in the second harvest year (Figure 8).

**Table 2.** Yield parameters (first-class and second-class marketable yield), as affected by planting densities.

| | First-Class Yield (g Plant$^{-1}$) | | Small Fruits (g Plant$^{-1}$) | | Misshapen Fruits (g Plant$^{-1}$) | | Rotten Fruits (g Plant$^{-1}$) | |
|---|---|---|---|---|---|---|---|---|
| **Plant density level (D)** | | | | | | | | |
| 30,000 | 394.60 ± 61.18 [1] | A | 39.93 ± 4.64 | A | 50.18 ± 0.02 | A | 1.94 ± 0.56 | A |
| 45,000 | 336.95 ± 45.23 | AB | 39.61 ± 7.55 | A | 45.36 ± 0.73 | A | 2.71 ± 0.14 | A |
| 60,000 | 266.74 ± 23.06 | BC | 42.20 ± 2.33 | A | 45.99 ± 2.97 | A | 1.89 ± 0.64 | A |
| 90,000 | 232.65 ± 21.67 | C | 42.83 ± 1.31 | A | 34.14 ± 0.17 | B | 1.26 ± 0.55 | A |
| 100,000 | 211.74 ± 17.33 | C | 32.03 ± 1.18 | A | 31.09 ± 5.23 | B | 1.96 ± 0.50 | A |
| *Significance* | *** | | ns | | *** | | ns | |
| **Year (Y)** | | | | | | | | |
| 2020 | 240.89 ± 24.98 | | 44.13 ± 3.11 | | 43.92 ± 3.58 | | 2.63 ± 0.17 | |
| 2021 | 336.18 ± 51.15 | | 34.51 ± 2.63 | | 38.79 ± 5.14 | | 1.28 ± 0.38 | |
| *Significance* | *** | | ** | | ns | | ** | |
| D × Y | ns | | ns | | ns | | ns | |

[1] Means ± SD (*n* = 4) followed by the same letter do not significantly differ according to LSD Fisher's test; $p < 0.05$. Two-way ANOVA significant differences: *** $p < 0.001$; ** $p < 0.01$; ns: not significant.

In both cropping seasons (2020 and 2021), the mean fruit weight was significantly higher (around +10%) in plants subjected to low planting densities (30,000 and 45,000) (Figure 9).

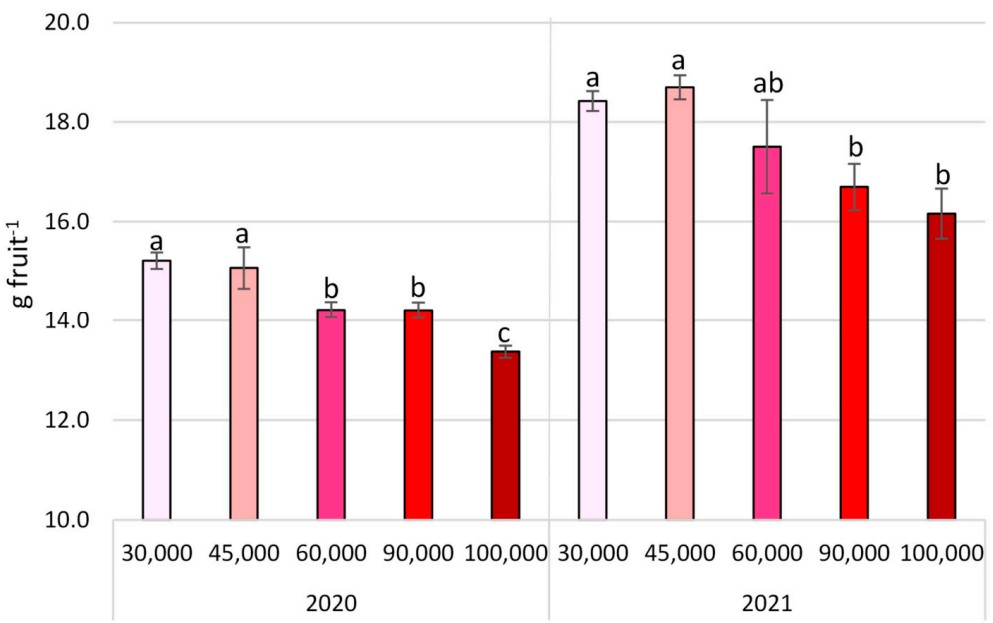

**Figure 9.** Average fruit weight (g fruit$^{-1}$) during the first (2020) and second (2021) cropping year, as affected by planting densities. Vertical bars indicate means ± SD (*n* = 4). Within each year, the letters indicate significant differences according to LSD Fisher's test; $p < 0.05$ (ns: not significant).

*3.3. Fruit Quality*

Strawberry qualitative traits assessed as flesh firmness (FF), total soluble solid (TSS), titratable acidity (TA), and color index (CI) were partially affected by plant density treatments (Table 3). As for FF, its average values were found higher in 2020 than in 2021, whereas the plant density was ineffective on this parameter. No change on TSS and TA was induced by different plant densities, as well as by the factor "year". Plant density had a visible and significant effect on CI, independently from the considered year. CI of fruits presented values ranging from 34 to 40, from light red to red, respectively. The highest CI value was observed in fruits coming from plants cultivated in wide spacing.

**Table 3.** Fruit quality traits (firmness (FF); total soluble solid (TSS); titratable acidity (TA); and color index (CI)), as affected by planting densities.

| | FF (Durofel Index) | | TSS (°Brix) | | TA (g Acid Citric L$^{-1}$) | | CI | |
|---|---|---|---|---|---|---|---|---|
| **Plant density level (D)** | | | | | | | | |
| 30,000 | 36.66 ± 4.87 [1] | A | 7.39 ± 0.09 | A | 6.67 ± 0.03 | A | 39.64 ± 1.78 | A |
| 45,000 | 36.19 ± 2.48 | A | 7.23 ± 0.12 | A | 6.50 ± 0.15 | A | 38.35 ± 1.10 | AB |
| 60,000 | 35.73 ± 2.63 | A | 7.04 ± 0.04 | A | 6.36 ± 0.34 | A | 36.12 ± 2.20 | ABC |
| 90,000 | 36.23 ± 2.81 | A | 7.09 ± 0.01 | A | 6.64 ± 0.04 | A | 35.86 ± 1.33 | BC |
| 100,000 | 35.87 ± 3.47 | A | 7.15 ± 0.09 | A | 6.85 ± 0.07 | A | 33.85 ± 1.82 | C |
| *Significance* | ns | | ns | | ns | | *** | |
| **Year (Y)** | | | | | | | | |
| 2020 | 40.74 ± 0.83 | | 7.15 ± 0.05 | | 6.77 ± 0.06 | | 34.44 ± 1.25 | |
| 2021 | 31.54 ± 0.59 | | 7.21 ± 0.12 | | 6.44 ± 0.18 | | 39.09 ± 1.09 | |
| *Significance* | *** | | ns | | ns | | *** | |
| **D × Y** | ns | | ns | | ns | | ns | |

[1] Means ± SD (*n* = 4) followed by the same letter do not significantly differ according to LSD Fisher's test; *p* < 0.05. Two-way ANOVA significant differences: *** *p* < 0.001; ns: not significant.

### 3.4. Economic Analysis

A synthesis of cost–benefit analysis is presented in the Figure 10. The total revenue (EUR ha$^{-1}$ over 2 cropping years), estimated from multiplying the produced strawberries by the local market price, increased linearly with decreasing the distance between plants (from 30,000 to 90,000). Beyond the planting density of 90,000, there was no increase from an economic point of view. Furthermore, the Figure 10 shows that a narrow plant spacing required a large investment (total costs). Among the tested density treatments, the two extremes (30,000 and 100,000) appeared economically unaffordable with regards to the net farm profit (EUR ha$^{-1}$ over 2 years). A similar farm profit was obtained adopting the planting densities of 60,000 and 90,000, while the maximum farm profit was reached with a density of 45,000 which corresponded to EUR 22,579 ha$^{-1}$ (over 2 years).

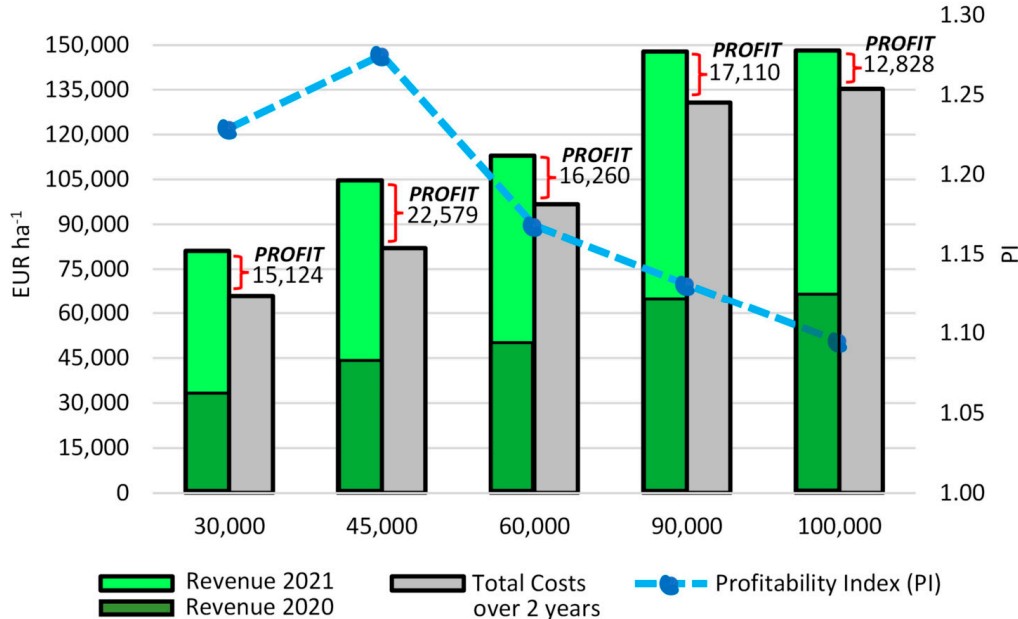

**Figure 10.** Revenue (green column: dark green = year 2020 and light green = year 2021), total costs (gray column; over 2 years), and net profit (values next to the curly brace; over 2 years), as affected by planting densities. Data are expressed as EUR ha$^{-1}$. The dashed blue line represents the profitability index (PI), as affected by planting densities.

Considering the profitability index (dashed blue line in the Figure 10), all planting density variants showed a value greater than 1. In particular, the widely spaced variants recorded the highest value (around 1.25).

## 4. Discussion

A severe intraspecific interaction is caused by an excessive presence of individuals belonging to the same species in a given space, which compete for key input resources [40,41]. Light, nutrients, and water can be considered the three main resource groups [42]. Competition between individual plants occurs primarily in the soil [43]. Root competition intensity increases as resource availability in soil decreases due to resource depletion or mechanisms of allelopathic interactions [44]. Although much research has indicated the correlation between the increase in root biomass accumulation with the decreasing planting density [45], our study did not highlight this relationship, but the increased plant biomass in low density system was due to the aerial part (Figure 2). Poor light exposure for leaves increases the aboveground competition intensity which can diminish with adequate light supply [46]. A large spacing between individual plants allows to maximize the light interception, avoiding negative effects that mutual and self-shading can have [19]. Indeed, our strawberry plants grown with the widest spacing were characterized by leaves with the greatest photosynthetic activity (+12% compared with plants in high planting density) (Figure 5). In addition to plant photosynthesis, light triggers many other essential physiological processes, for example, flower bud initiation and branching, as reported by Yang and Jeong [47] in chrysanthemum plants. Our study confirmed that more flower trusses, flowers, and branch crowns per plants were found in plants more exposed to light (i.e., in low density systems) (Figure 6; Figure 4). However, these findings appeared only during the second cropping year. Auxiliary buds form branch crowns and the apical meristems of crowns develop into terminal inflorescences in autumn, depending on environmental conditions such as daylength and temperature [23,48,49]. Since our plant materials came from the same nursery and the induction/differentiation phase took place under standard conditions for all plants, no significant differences were observed in the reproductive aspects during the first cropping year. On the contrary, subjecting plants to different growing conditions (i.e., planting densities) in autumn 2020 (i.e., at the end of first harvesting year) helped induce morphological changes that would have been evident the following year (2021). Considering this explanation, the significant variations in total and commercial yield in the first harvesting year must be found not in the number of flowers per plant (instead, for the second cropping year) but in the average fruit weight (Table 2; Figure 9). Plant–plant interaction for limited resources can lead to differentiated investment in their growth and reproduction [50]. Strawberry plants under nutrient limitation responded by favoring the vegetative growth (i.e., leaf and root biomass) to the detriment of the plant's reproductive investment, as evidenced by the low yield and very small size of the fruits, observed by Soppelsa et al. [51].

Picking productivity of open-field strawberries (e.g., cultivar Elsanta) for the fresh market is usually at 12–15 kg per hour [52]. In the cultivation area where we carried out the trial, the productivity during harvesting is rather around 10–12 kg per hour (or less), for the same variety. This lower picking productivity can be associated with hostile growing conditions. For instance, the sloping land plots in Martell Valley slow down harvesting operations. Moreover, the exceptional environmental conditions typical of this valley affect pomological traits such as fruit weight. Indeed, Naryal et al. [53] reported that the apricot fruit weight decreases by 0.5 g for every 100 m of increase in altitude. Since in our study a significant difference emerged for the fruit weight parameter (i.e., greater fruit size in low planting density), we took this aspect into account to calculate a differentiated picking productivity in the economic analysis (Figure 10).

As reported in Figures 8 and 10, the increased yield per area with a high plant density (90,000 or 100,000 plant ha$^{-1}$) led to an increase in total revenue (EUR ha$^{-1}$) but the total costs (EUR ha$^{-1}$) also reached a considerably high level. We need an investment cost of

EUR 75,000 per hectare with a low planting density, while more than EUR 130,000 are necessary with a high density, which means an increase of about EUR 60,000 (i.e., +80%). Harvesting costs account for more than 40% of total business costs. Similar percentage among the various planting densities tested. Another important item is represented by the cost of the plants (EUR 0.332 per plant + vat), which varies among the densities tested due to the different number of plants transplanted.

High revenue is not synonymous with high profit, since costs also increase proportionally. Therefore, the profit-maximizing planting spacing was achieved with 45,000 plants ha$^{-1}$. The result of our study is consistent with the findings by Matsumoto et al. [54] and Castellanos et al. [55], who state that a middle planting density (or better to say not too high) in upland rice or garlic cultivation is preferable for the highest farm profit. Furthermore, the choice of the right planting density has a noticeable influence on opportunity costs, as reported by Jettner et al. [56] comparing different sowing rates of faba bean (*Vicia faba* L.).

Under the described growing conditions, the different plant densities had no significant effect on the main fruit quality traits, except for the color index. With the decrease in the spacing between individual plants, there is less sunlight exposure for fruits, which affects their coloration (Table 3). These findings are confirmed by Martins de Lima et al. [16]. Sunlight has a remarkable effect on regulating the biosynthesis of phenolic compounds such as anthocyanins, which is the major pigment in strawberry fruits [57,58]. Having more intense red fruits sometimes reflects the ideal purchase intention of the consumer [59].

As mentioned before, not only can cultural practices affect availability of resources but environmental factors (e.g., elevation) can also play an important role [60]. For example, high-altitudinal levels were observed to change physiological and morphological responses of plants, interfering precisely with resources such as radiation [35]. The positive influence of high elevation on strawberry nursery materials was described by Maroto et al. [61] and Pirlak et al. [62], showing an increased number of leaves, runners, and flowers in that cultivation condition. Although the effects of an agronomic practice such as planting density has been well-investigated on different vegetable and fruit crops, including strawberry [10,11,14,32,63], no previous research has been conducted under our imposed experimental conditions (i.e., in alpine mountain environment).

## 5. Conclusions

Optimizing the planting density is an effective strategy for improving yield and farm profit, especially in alpine mountain environments. Given the results, as summarized in Figure 11, recommendations from this study are: not to exceed the density of 100,000 plants ha$^{-1}$ (economically disadvantageous); adopt medium or low planting densities to have strong plants (e.g., for a third year of continuation); pay attention to correctly manage the picking times with low planting density in order to avoid overripe fruits with a dark intense red color; having fewer plants per hectare reduces the total costs but can increase the business risk (e.g., loss of plants due to crown and root rot, loss of production due to strawberry blossom weevil (*Anthonomus rubi*), etc.); and to encourage producers to adopt wide spacing between plants for more sustainable strawberry growing (we observed a lower incidence of powdery mildew on plants subjected to a low planting density). Further research is needed to examine the agronomic and economic benefits of influencing planting density in a soilless cultivation system for strawberries, a cultivation technique that is increasingly gaining popularity in South Tyrol.

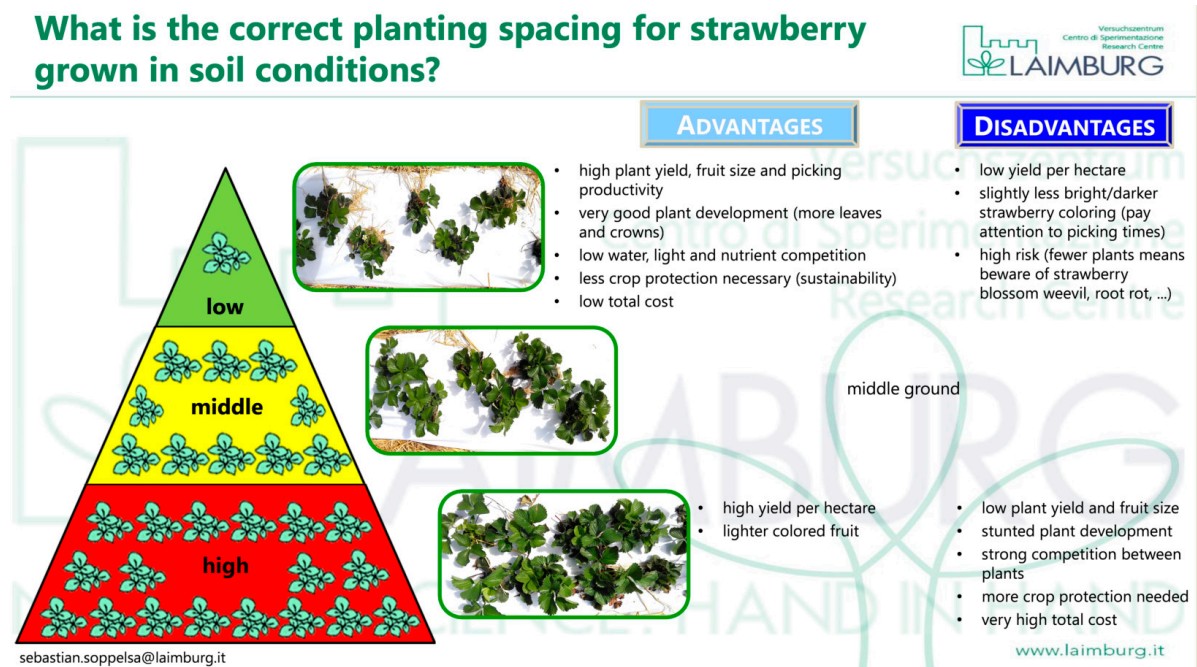

**Figure 11.** Schematic representation of advantages and disadvantages by adopting different planting densities in strawberry soil cultivation.

**Author Contributions:** S.S. and M.Z. conceived and designed the experiment; S.S. and M.G. performed the experiment; S.S. analyzed the data; S.S. and M.G. wrote the manuscript. All authors have read and agreed to the published version of the manuscript.

**Funding:** This research activity was conducted in the framework of the "Action Plan for Mountain Agriculture and Food Sciences 2016–2022, adopted by the Government of South Tyrol (Italy).

**Institutional Review Board Statement:** Not applicable.

**Informed Consent Statement:** Not applicable.

**Data Availability Statement:** Not applicable.

**Acknowledgments:** Preliminary data from this research study earned the author S.S. the ISHS Young Minds Award during the IX International Strawberry Symposium. S.S. wishes to thank the Symposium Commission for having given him this important recognition. The authors would like to thank Peter Gamper (former mayor of Martello/Martell) for his support during the field activity and for the helpful suggestions regarding the cost–benefit analysis. The authors thank the Department of Innovation, Research University and Museums of the Autonomous Province of Bozen/Bolzano for covering the Open Access publication costs.

**Conflicts of Interest:** The authors declare no conflict of interest.

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
