# Peer review of "Optimizing Planting Density in Alpine Mountain Strawberry Cultivation in Martell Valley, Italy"

_agronomy, doi:10.3390/agronomy13051422_

Round 1

Reviewer 1 Report

The study is comprehensive and important and is planned from a different perspective. I thank the researchers for their labor-intensive work. However, by making the following correction suggestions, the study will be more understandable and will be more beneficial for researchers working on this subject.

The title is not appropriate. Country and region should not be written in parentheses.

In abstract part:

The summary should be reviewed and written with the support of statistical analysis. The summary part should be enriched with mostly numerical data and statistical significance levels.

Line 14 95.000 written, but 90.000 in text and tables. correct it and edit if there are similar errors in the text.

The number of plants per hectare was given, but the results were interpreted by keeping the planting intervals wide.

It should be clarified which is wide and middle plant spacing.

add some imformation about gas exchange parameters and Economic analysis in this part

"elevation and flowering" in keywords are not suitable. Ä°nstead of elevation I think you should use "altitude"  and even the altitude of the region should be given in the summary or text.

Introduction part

Line 64 "cold-stored" In this section, did you mention cold-resistant varieties or cold storage?

Line 105-109 Why give an example of cotton? It would be more appropriate to give examples from the studies carried out on strawberries. If there is no study on this subject in strawberries, you can give an example of vegetables.

Line 121 delete "like that in the Martell  Valley." I know this comparison is very important for you, but highlighting this important information in the summary and discussion sections will increase the value of your work.

Materials and Methods part

Line 213 "photosynthetic data"  did you mean gas exchange parameters. edit this part in the material method and discussion section

Author Response

We thank the reviewer for their careful reading of the paper and their positive comments. We welcome the reviewer's suggestions and have considered each of these in the revised version of the manuscript (see attachment).

Reviewer 2 Report

The authors tried to optimize planting density in Martell Valley.

However, there are major issues.

1. All the results are predictable.

2. I don't understand why the treatments are set to 30,000, 45,000, 60,000, 90,000, and 100,000 plant/ha-1.

3. Hard to read the English. Moreover, the style is not for scientific manuscript (For instance, we don't use cm-1 because cm is not SI unit.

4. The number of samples is not enough (4 samples).

5. In the Fig. 10, the two years results should be displayed separately.

6. Environmental data is not enough to demonstrate the results.

7. In the Fig. 11, the triangle is inappropriate to say planting spacing.

My comments are in the file.

Thanks,

Author Response

We thank the reviewer for their careful reading of the paper and their comments. We welcome the reviewer's suggestions and have considered each of these in the revised version of the manuscript (see attachment).

Round 2

Reviewer 2 Report

I just found minor issues.

The aboveground should be shoot.

The belowground should be root.

The temperature should be air temperature (Table 1).

Thanks,

Author Response

Response to Reviewer 2 Comments

  • The aboveground should be shoot.

We cannot change this term. In support of our decision, we indicate below the links of publications in which the term "aboveground biomass" is used for strawberry plant.

https://doi.org/10.3389/fpls.2021.700479

in M&M: “The aboveground biomass [fresh weight (FW) and dry weight (DW, 48 h at 70°C)] and the belowground root system were weighed after sampling ….”

https://doi.org/10.3390/agronomy12092082

in Results: “… mainly due to significant differences in aboveground biomass production, …”

https://doi.org/10.1007/s12550-022-00451-5

https://doi.org/10.21273/JASHS.125.3.324

https://www.jstor.org/stable/26505870

https://doi.org/10.1016/j.scienta.2021.110579

  • The belowground should be root.

Ok, done!

  • The temperature should be air temperature (Table 1).

Ok, done!